

# Antimicrobial activity of *Streptomyces* spp. isolated from *Apis dorsata* combs against some phytopathogenic bacteria

Yaowanoot Promnuan[1], Saran Promsai[1] and Sujinan Meelai[2]

[1] Department of Microbiology, Faculty of Liberal Arts and Science, Kasetsart University-Kamphaeng Saen campus, Kamphaeng Saen, Nakhon Pathom, Thailand
[2] Department of Microbiology, Faculty of Science, Silpakorn University-Sanam Chandra Palace campus, Nakhon Pathom, Nakhon Pathom, Thailand

## ABSTRACT

The aim of this study was to investigate the antimicrobial potential of actinomycetes isolated from combs of the giant honey bee, *Apis dorsata*. In total, 25 isolates were obtained from three different media and were screened for antimicrobial activity against four plant pathogenic bacteria (*Ralstonia solanacearum*, *Xanthomonas campestris* pv. *campestris*, *Xanthomonas oryzae* pv. *oryzae* and *Pectobacterium carotovorum*). Following screening using a cross-streaking method, three isolates showed the potential to inhibit the growth of plant pathogenic bacteria. Based on a 96-well microtiter assay, the crude extract of DSC3-6 had minimum inhibitory concentration (MIC) values against *X. oryzae* pv. *oryzae*, *X. campestris* pv. *campestris*, *R. solanacearum* and *P. carotovorum* of 16, 32, 32 and 64 mg $L^{-1}$, respectively. The crude extract of DGA3-20 had MIC values against *X. oryzae* pv. *oryzae*, *X. campestris* pv. *campestris*, *R. solanacearum* and *P. carotovorum* of 32, 32, 32 and 64 mg $L^{-1}$, respectively. The crude extract of DGA8-3 at 32 mg$L^{-1}$ inhibited the growth of *X. oryzae* pv. *oryzae*, *X. campestris* pv. *campestris*, *R. solanacearum* and *P. carotovorum*. Based on their 16S rRNA gene sequences, all isolates were identified as members of the genus *Streptomyces*. The analysis of 16S rRNA gene sequence similarity and of the phylogenetic tree based on the maximum likelihood algorithm showed that isolates DSC3-6, DGA3-20 and DGA8-3 were closely related to *Streptomyces ramulosus* (99.42%), *Streptomyces axinellae* (99.70%) and *Streptomyces drozdowiczii* (99.71%), respectively. This was the first report on antibacterial activity against phytopathogenic bacteria from actinomycetes isolated from the giant honey bee.

Corresponding author
Yaowanoot Promnuan,
faasynp@ku.ac.th

## INTRODUCTION

The Gram-negative bacteria *Xanthomonas campestris* pv. *campestris*, *Xanthomonas oryzae* pv. *oryzae*, *Ralstonia solanacearum* and *Pectobacterium carotovorum* are known to cause significant losses in many crop plants worldwide. *X. campestris* pv. *campestris* is a seed-borne pathogen that causes black rot disease in a large number of species of the Brassicaceae, including the genera *Brassica* and *Arabidopsis*. The typical disease symptoms include V-shaped yellow lesions starting from the leaf margins and blackening of the veins

(*Vicente & Holub, 2013*). *X. oryzae* pv. *oryzae* causes devastating bacterial bright leaf (BLB) disease, which is one of the major diseases of rice in Asian countries. This bacterial pathogen grows in the xylem vessel, causing yellow/white lesions along the leaf veins (*Xie et al., 2018*). *R. solanacearum* is the causal agent of bacterial wilt, which is one of the most devastating plant diseases worldwide. This soil-borne vascular pathogen can cause disease to many economically important crops, including tomato, potato, eggplant, tobacco and banana. The bacterium infects plants via wounds or root tips; it invades the xylem vessels and systematically spreads to the aerial parts of the plant through the vascular system (*Ombiro et al., 2018*). *P. carotovorum*, formerly known as *Erwinia carotovora*, is one of the most destructive diseases of postharvest vegetables worldwide, especially potatoes, green peppers and Chinese cabbages (*Zhao et al., 2013*). The bacterium is found on plant surfaces and in soil, where it may enter the plant via wound sites or through natural openings on the plant surface. Once inside the plant, it resides in the vascular tissue and intracellular spaces, where it remains until environmental conditions, including free water, oxygen availability and temperature, become suitable for disease development (*Itoh et al., 2003*). Soft rot pathogens cause general tissue maceration, termed soft rot disease, through the production of enzymes that degrade plant cell walls.

The management of plant diseases is difficult. The use of resistant cultivars to control bacterial wilt disease is the most economical, environmentally friendly and effective method (*Ombiro et al., 2018*). The disease control of black rot relies on the use of pathogen-free seed and planting material and the elimination of other potential inoculum sources (*Vicente & Holub, 2013*). Currently, the control of postharvest bacterial soft rot depends mainly upon the use of bactericides, such as hypochlorite, formaldehyde solution and antibiotics (*Zhao et al., 2013*). However, the use of chemical bactericides and antibiotics to control phytopathogenic bacteria could cause serious damage to the environment and human health. Moreover, some emerging strains have shown strong resistance to all these products (*Sabir et al., 2017*; *Mougou & Boughalleb-M'hamdi, 2018*; *Wu et al., 2019*). Therefore, many researchers have focused on the development of alternative methods of controlling plant diseases. The use of antibacterial compounds from plant extracts (*Satish, Raveesha & Jandrdhana, 1999*; *Kaur et al., 2016*), validamycin A (*Ishikawa et al., 2004*), xantho-oligosaccharide (*Qian et al., 2006*) and ralhibitins (*Ombiro et al., 2018*) to inhibit the growth of phytopathogens has been studied and reported. There have been several studies of antagonistic microorganisms, such as *Bacillus* spp. and *Pseudomonas* spp., endophytic actinomycetes and melanogenic actinomycetes to inhibit the growth of bacteria causing black rot and bacterial bright leaf disease (*Wulff et al., 2002*; *Mishra & Arora, 2011*; *Zhao et al., 2013*; *Muangham, Pathom-aree & Duangmal, 2015*). Several *Streptomyces* species, such as *Streptomyces aureofaciens*, *Streptomyces avermitilis*, *Streptomyces humidus*, *Streptomyces hygroscopicus*, *Streptomyces lividans*, *Streptomyces lydicus*, *Streptomyces olivaceoviridis*, *Streptomyces plicatus*, *Streptomyces roseoflavus*, *Streptomyces scabies* and *Streptomyces violaceusniger*, have been used to control soil-borne diseases for their intense antagonistic activities by the production of various antimicrobial substances (*Zheng et al., 2019*).

Actinomycetes, especially *Streptomyces* species, are well known for producing bioactive compounds which suggests that actinobacteria have the potential to produce antimicrobial compounds against phytobacterial pathogens (*Viaene et al., 2016*). Actinomycetes isolated from different habitats have been investigated for antimicrobial activities against plant pathogens; for example, melanogenic *Streptomyces* isolated from rhizospheric soils had the ability to inhibit the growth of rice pathogenic bacteria (*X. oryzae* pv. *oryzae* and *X. oryzae* pv. *oryzicola*). Among these, isolate TY68-3 had the highest antibacterial activity and siderophore production and had 99.6% 16S rDNA sequence similarity to *S. indiaensis* (*Muangham, Pathom-aree & Duangmal, 2015*). In addition, *S. caeruleatus* isolated from the rhizosphere soil of *Cassia fistula* had the highest activity against the soybean pathogen *X. campestris* pv. *glycine* (*Mingma et al., 2014*). *Hastuti et al. (2012)* reported that endophytic *Streptomyces* reduced *X. oryzae* pv. *oryzae* infection in rice. Some isolates were able to improve the growth of rice seedlings, plant height and dry weight. In addition, *Streptomyces* strain LBR02 had the highest inhibitory activity (25 mm diameter inhibition zone) against *X. oryzae* pv. *oryzae in vitro*. Furthermore, *S. violaceusnige* (strain A5) was isolated from chitin-rich partially decomposed molted snakeskin and had maximum inhibitory activity (0.625–1.25 mg mL$^{-1}$) against *X. axonopodis* pv. *punicae*, the causative agent of oily spot disease in pomegranate (*Chavan et al., 2016*). These reports indicated that actinomycetes, especially *Streptomyces*, may provide a new approach for the use of actinomycetes for biocontrol in agriculture.

The giant honey bees, consisting of the species *Apis dorsata*, *Apis laboriosa* and *Apis breviligula*, are distributed over a vast geographic area in South and Southeast Asia. In Thailand, only *A. dorsata* is found. Bees of this species build a massive single comb attached under the surface of a stout tree branch or an overhang of a rock face, or sometimes to the eves of buildings or other urban structures (*Wongsiri et al., 1996*). The actinomycetes associated with *A. dorsata* have never been studied or reported. However, there have been several studies of actinomycetes associated with bees and stingless bees in Thailand. The novel actinomycete species *Actinomadura apis* was isolated from the honey bee (*Apis mellifera*) (*Promnuan, Kudo & Chantawannakul, 2011*). Two novel species of the genus *Streptomyces* (*Streptomyces chiangmaiensis* and *Streptomyces lannensis*) were isolated from stingless bee (*Tetragonilla collina*) collected from Chiang Mai Province, northern Thailand (*Promnuan, Kudo & Chantawannakul, 2013*). Thirty-two actinobacteria isolates were obtained from honey bees (*A. mellifera*, *Apis cereana* and *Apis florea*). Most of the isolates belonged to the genus *Streptomyces*. Some less frequent isolates were classified in the genera *Nonomuraea*, *Nocardiopsis* and *Actinomadura*. Moreover, some of these isolates produced antimicrobial compounds that inhibited the growth of the honey bee pathogens *Paenibacillus larvae* and *Melisococcus plutonius*, which cause American and European foulbrood diseases in honey bees, respectively (*Promnuan, Kudo & Chantawannakul, 2009*). These studies indicated that actinomycetes associated with bees have the potential to produce antimicrobial compounds to combat disease in agriculture. However, the rate of discovery of new antibiotics from actinomycetes from common habitats has slowed down; therefore, novel antibiotics must be found from actinomycetes in unexplored habitats (*Berdy, 2005*).

This study focused on actinomycetes isolated from *A. dorsata* combs and evaluated their antibacterial activity against plant pathogenic bacteria. According to data obtained in the current study, the antibacterial activity of metabolites from actinomycetes could be implemented against phytopathogenic bacteria to assist in crop protection.

## MATERIALS & METHODS

### Sample collection

Three combs of the giant honey bee (*A. dorsata*) were collected from the Mae-rim district, Chiang Mai Province, Thailand in April 2014. The three hive samples were collected from local villages in private areas. Verbal permission was acquired from each of the owners: 1. Mr. Ton Tatiya (2 hives) 25/2 Moo2, Pong Yaeng Sub-district, Mae-Rim district, Chiang Mai Province, Thailand. 2. Mr. Ma Madamun (1 hive) 93/3 Moo2, Pong Yaeng Sub-district, Mae-Rim district, Chiang Mai Province, Thailand. Adult bees, pollen and honey were collected and kept in sterile tubes and stored at −20 °C until the isolation process.

### Actinomycete isolation

Three adult bees were surface-sterilized using a triple surface-sterilization technique and ground aseptically following the method modified from *Photita et al. (2004)* and *Promnuan, Kudo & Chantawannakul (2009)*. Isolation of actinomycetes from pollen and honey was obtained using a standard dilution plate method on starch casein nitrate agar (*Küster & Williams, 1964*), glycerol-asparagine (ISP5) (*Pridham & Lyons, 1961*) and Czapek's agar (*Waksman, 1950*) supplemented with 25 $\mu$g mL$^{-1}$ nystatin and nalidixic acid. Plates were incubated at 30 °C for 7–21 days and examined periodically. The actinobacterial colonies were isolated, purified and maintained in yeast extract-malt extract agar (ISP2) (*Shirling & Gottlieb, 1966*) slants and stored at 4 °C.

### Test organisms

*X. campestris* pv. *campestris*, *X. oryzae* pv. *oryzae*, *R. solanacearum* and *P. carotovorum* were obtained from the Department of Agriculture, Ministry of Agriculture and Cooperative, Thailand. The phytobacterial pathogens were activated and maintained in nutrient agar (NA) and ISP2 agar for 24–48 h at 30 °C before use.

### Screening of antagonistic actinomycetes

In total, 25 actinomycete isolates were evaluated for their activity toward four plant pathogenic bacteria: *X. campestris* pv. *campestris, X. oryzae* pv. *oryzae, R. solanacearum* and *P. carotovorum* using a modified cross streak method (*Lemos, Toranzo & Barja, 1985*). Each isolate was streaked on the center of ISP2 and a glucose yeast extract (GYE) (*Agate & Bhat, 1963*) agar plate and incubated at 30 °C for 7 days. Each test organism was streaked across the actinomycete line and incubated at 30 °C for 24 h. Then, the inhibition zones were observed and measured. The experiment was conducted in triplicate.

### Extraction of bioactive compounds

The actinobacterial isolates that showed potent activity against the growth of the test organisms in the previous method were grown on ISP2 agar plates and incubated at

30 °C for 14 days. The metabolites were extracted using a modified extraction method as described by *Kumar et al. (2012)*. The culture medium of each isolate was cut into small pieces (approximately 0.5 × 0.5 cm), extracted with 200 mL of ethyl acetate, shaken vigorously on a flask shaker at 150 rpm and 30 °C for 48 h and then filtered using filter paper (Whatman No.1). The ethyl acetate fractions were concentrated using a rotary vacuum evaporator at 40 °C. The crude extracts were resuspended in 1 mL of sterile dimethyl sulfoxide (DMSO) and stored at −20 °C until testing for antimicrobial activity.

## Determination of the minimum inhibitory concentration (MIC) of selected actinomycetes

The MIC of crude extract was determined according to *Wiegand, Hilpert & Hancock (2008)* with some modifications. The concentrations of the test organisms, *X. campestris* pv. *campestris, X. oryzae* pv. *oryzae, R. solanacearum* and *P. carotovorum*, were adjusted to the equivalent of 0.5 McFarland standard. The crude extract was dissolved in sterile DMSO to obtain an initial concentration of 5,120 mg L$^{-1}$. Sterile DMSO was used as a negative control. The MIC of each extract was determined using serial two-fold dilutions in ISP2 broth in a 96-well microtiter plate (concentration range from 0.5 to 256 mg L$^{-1}$). The experiment was performed in triplicate. Each microtiter plate was incubated at 30 °C for 24 h. After incubation, the suspension from each well was streaked onto separate ISP2 agar plates and incubated at 30 °C for 24 h, after which any growth was observed. The minimum concentrations of the extracts that showed no turbidity and had no bacterial growth on the ISP2 agar plate were recorded as the MIC values.

## Identification of actinomycetes using 16S rDNA sequencing

The selected isolates that showed potent activity against the test organisms were grown in 50 mL of ISP2 broth and incubated in a shaker (120 revolutions min$^{-1}$) at 30 °C for 7 days. Then, cells were collected via centrifugation (91,000 g) for 5 min and were washed three times using sterile distilled water. Genomic DNA was extracted, and the 16S rRNA gene was amplified using the methods described by *Nakajima et al. (1999)*. The primers used for amplification were 20F (5′-AGTTTGATCCTGGCTC) and 1540R (5′-AAGGAGGTGATCCAGCC). Then, PCR products were purified using an Invitrogen$^{TM}$ PureLink$^{TM}$ PCR Purification Kit (Thermo Fisher Scientific, USA) according to the manufacturer's instructions. Purified PCR products were sequenced by the Sanger method at 1st BASE, Singapore. The highest similarity of actinomycetes with the reference species was confirmed using the NCBI BLAST tool (https://blast.ncbi.nlm.nih.gov/Blast.cgi). The sequences of closely related type strains retrieved from the GenBank database were multiple aligned using Clustal_W in BioEdit Sequence Alignment Editor 7.2.5 (*Hall, 1999*). After multiple alignments, a phylogenetic tree was constructed using the maximum likelihood (ML) method in MEGA X version 10.1.8 (*Kumar et al., 2018*) based on a comparison of 1,332–1,374 nucleotides present in all the strains used after elimination of gaps and ambiguous nucleotides from the sequences. *Streptomyces thermocarboxydus* DSM 44293$^{T}$ was used as an outgroup. Confidence values for branches of the phylogenetic tree were determined using bootstrap analyses based on 1,000 resamplings (*Felsenstein, 1985*). The

**Table 1** Numbers of actinomycetes isolated from *A. dorsata* using glycerol asparagine agar (ISP5), starch casein nitrate agar (SC) and Czapek's agar (CZ).

| Sample | Isolation medium | | | Total (%) |
|---|---|---|---|---|
| | ISP5 | SC | CZ | |
| Adults | 1 | 0 | 0 | **1 (4%)** |
| Pollen | 11 | 5 | 5 | **21 (84%)** |
| Honey | 3 | 0 | 0 | **3 (12%)** |
| **Total (%)** | **15 (60%)** | **5 (20%)** | **5 (20%)** | **25 (100%)** |

sequence similarity values were calculated from the pairwise alignments obtained using BioEdit 7.2.5 (*Hall, 1999*).

## RESULTS

### Isolation of actinomycetes from *A. dorsata*

The samples (adult bees, pollen and honey) were collected from three hives of *A. dorsata*. Twenty-five morphologically different actinobacterial isolates were obtained from three different media, with 60% from ISP5 agar followed by starch casein nitrate agar and Czapek's agar. Most of the actinomycetes were isolated from pollen (84%), followed by honey (12%) and adult bees (4%) (Table 1).

### Antimicrobial activity against plant pathogens

Based on screening for antimicrobial activity using the cross-streaking method, three actinomycete isolates (DSC3-6, DGA3-20 and DGA8-3) showed potent activity (>10 mm diameter inhibition zones) against the growth of four phytopathogenic bacteria (*X. campestris* pv. *campestris, X. oryzae* pv. *oryzae, R. solanacearum* and *P. carotovorum*) on ISP2 agar plates (Fig. 1). The crude extracts of the three isolates were subsequently tested for their MIC levels against the growth of phytopathogenic bacteria. Based on the 96-well microtiter assay, the MIC values of the crude extract of the three actinobacterial strains are shown in Table 2. The MIC value of actinobacterial strain DSC3-6 against *X. oryzae* pv. *oryzae* was 16 mg L$^{-1}$, and the MIC values of the crude extracts of DGA3-20 and DGA8-3 were both 32 mg L$^{-1}$. All isolates inhibited the growth of *X. campestris* pv. *campestris, R. solanacearum* and *P. carotovorum*, with MIC values of 32, 32 and 64 mg L$^{-1}$, respectively.

### Identification of actinomycetes using 16S rDNA

The three actinobacterial isolates that showed inhibition of all four phytobacterial pathogens were identified using 16S rDNA sequencing. 16S rRNA gene sequences for strains DSC3-6 (LC536753), DGA3-20 (LC536752) and DSC8-3 (LC536754) were analyzed by BLAST using the GenBank database. The results showed that all strains had high similarity to members of the genus *Streptomyces*. The almost complete 16S rRNA gene sequences for strains DSC3-6, DGA3-20 and DGA8-3 were compared with the corresponding sequences of closely related strains of the genus *Streptomyces*. The maximum likelihood tree (Fig. 2)

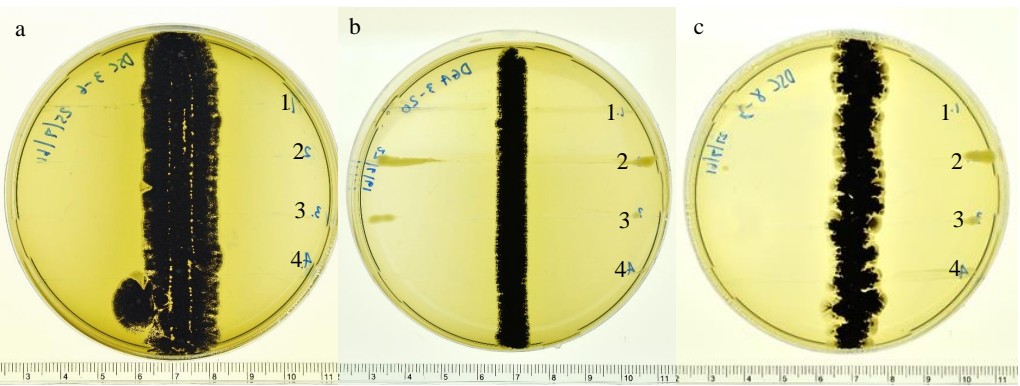

**Figure 1 Inhibitory effects of actinomycetes.** Inhibitory effects of actinomycetes against the growth of 1. *R. solanacearum*; 2. *X. campestris* pv. *campestris*; 3. *X. oryzae* pv. *oryzae* and 4. *P. carotovorum* on ISP2 agar plates: (A) DSC3-6, (B) DGA3-20 and (C) DGA8-3.

**Table 2 Characterization and identification of actinomycetes using the 16S rDNA gene sequence and minimum inhibitory concentration (MIC) values of actinomycetes against the growth of 1, *X. campestris* pv. *campestris*; 2, *X. oryzae* pv. *oryzae*; 3, *R. solanacearum* and 4, *P. carotovorum*.**

| Isolate No. | Morphological characteristic | Source | Accession No. | MIC (mg L⁻¹) | | | | 16S rDNA gene identification (% similarity) |
|---|---|---|---|---|---|---|---|---|
| | | | | 1 | 2 | 3 | 4 | |
| DSC3-6 | Powdery colonies<br>Substrate mycelium: cream<br>Aerial spore mass: grey or black<br>Produce yellow pigment | Pollen | LC536753 | 32 | 16 | 32 | 64 | *Streptomyces ramulosus* (99.42%) |
| DGA3-20 | Powdery colonies<br>Substrate mycelium: cream or grey<br>Aerial spore mass: grey or black | Pollen | LC536752 | 32 | 32 | 32 | 64 | *Streptomyces axinellae* (99.70%) |
| DGA8-3 | Powdery colonies<br>Substrate mycelium: cream<br>Aerial spore mass: grey or black | Honey | LC536754 | 32 | 32 | 32 | 64 | *Streptomyces drozdowiczii* (99.71%) |

revealed that strains DSC3-6, DGA3-20 and DGA8-3 were closely related to *S. ramulosus*, *S. axinellae* and *S. drozdowiczii*, respectively.

The sequence similarity value between each actinomycete isolate and its closely related type strain was aligned and calculated from the pairwise alignment. The results showed that DSC3-6, DGA3-20 and DGA8-3 were closely related to *S. ramulosus* (99.42%), *S. axinellae* (99.70%) and *S. drozdowiczii* (99.71%), respectively (Table 2).

## DISCUSSION

This study investigated antagonistic activity against phytopathogenic bacteria of actinomycetes isolated from the giant honey bee (*A. dorsata*). Using ISP5 agar, DGA3-20 and DGA8-3 were obtained from pollen and honey samples. However, isolate DSC3-6, which had the highest activity against *X. oryzae* pv. *oryzae*, was obtained from pollen

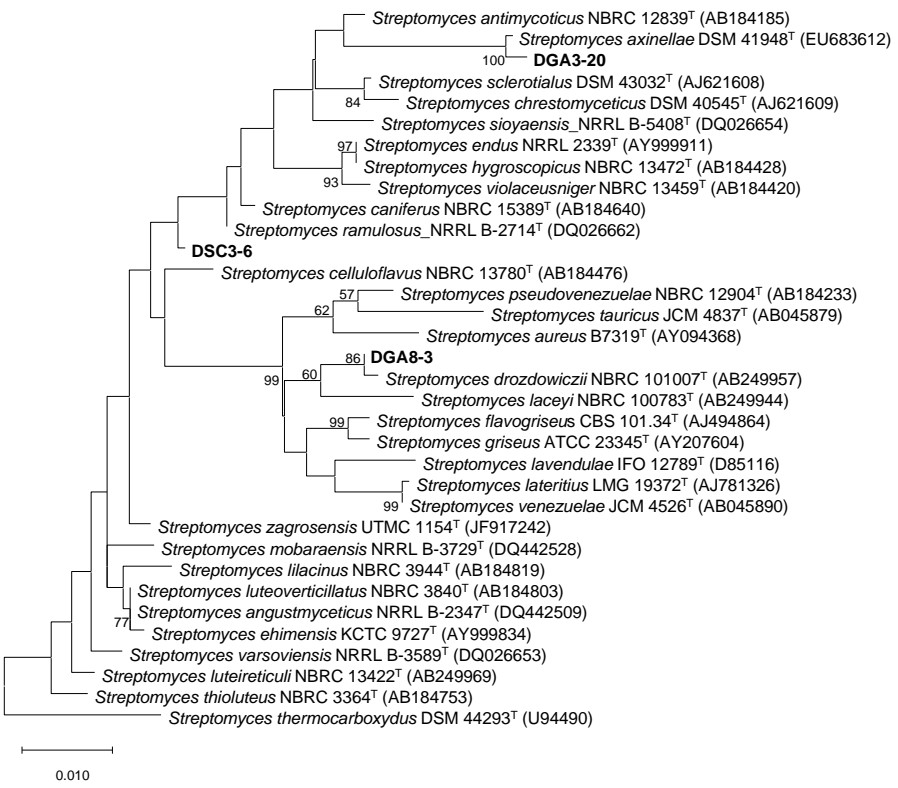

**Figure 2** **Maximum likelihood (ML) tree based on 16S rRNA gene sequences showing the phylogenetic positions of DSC3-6, DGA3-20 and DSC8-3 relative to type strains of other *Streptomyces* species.** *S. thermocarboxydus* DSM 44293[T] was used as an outgroup. The number at each node is the bootstrap support value (%) based on 1,000 replicates (only values > 50% are shown). The scale bar shows 0.010 substitutions per nucleotide position.

using starch casein nitrate agar. This indicated that using different isolation media may increase the opportunity of finding potential actinomycete strains. The three *Streptomyces* strains capable of inhibiting phytopathogens were obtained from pollen and honey stored in combs. The actinomycetes may be taken into hives by the worker bees collecting food and/or water from environmental sources outside the hives (*Promnuan, Kudo & Chantawannakul, 2009*). *Streptomyces* were isolated from strawberry flowers and pollen cultivated in high-bed greenhouses in Jinju, Republic of Korea. This study showed that honey bees (*A. mellifera*) can transfer *Streptomyces* bacteria among flowers and strawberry plants. In addition, these endophytic *Streptomyces* had the ability to protect both plant and honey bees from phytopathogenic fungi (*Botrytis cinerea*) and entomopathogens, respectively (*Kim et al., 2019*).

Based on 16S rDNA sequence analysis, all potent isolates belonged to the genus *Streptomyces*. Actinomycetes, especially *Streptomyces* are well known for the production of secondary metabolites with antagonistic activity against phytopathogens (*Viaene et al., 2016*). There have been reports of actinobacteria associated with insects. For example, *Streptomyces*, *Micromonospora* and *Actinoplanes* isolated from nests of the paper wasp

*Polistes dominulus* could inhibit the growth of *Pseudomonas aeruginosa*, *Escherichia coli*, *Staphylococcus aureus*, *Serratia marcescens* and *Bacillus subtilis* (*Madden et al., 2013*). The *Streptomyces* spp. isolated from solitary wasp mud nests showed activity against various drug-resistant bacterial pathogens. The isolate MN 9(V) showed activity against both *E. coli* and *P. aeruginosa* at a concentration of 25 mg mL$^{-1}$ (*Kumar et al., 2012*). The novel macrocyclic lactam sceliphrolactam was isolated from mud dauber wasps (*Chalybion californicum* and *Sceliphron caementarium*) and could act as an antifungal by destabilizing fungal cell membrane functions (*Poulsen et al., 2011*). Actinomycetes associated with bees (*A. mellifera*, *A. cereana* and *A. florea*) produce antimicrobial compounds that inhibit the growth of bacterial pathogens causing American and European foulbrood diseases in honey bees (*Promnuan, Kudo & Chantawannakul, 2009*). Furthermore, *Streptomyces* spp. isolated from black dwarf honey bee (*A. andreniformis*), showed high activity in decreasing the egg hatch rate and increasing the infective second-stage juvenile mortality rate of the root-knot nematode (*Meloidogyne incognita*) *in vitro* and reduced root gall of chili *in vivo* (*Santisuk et al., 2018*). These results indicated that actinomycetes associated with insects can provide novel antimicrobial products for use in agriculture.

## CONCLUSIONS

Bacterial bright leaf disease, which is caused by *X. oryzae* pv. *oryzae*, is one of the major diseases of rice in Asian countries. This study is the first report of the antibacterial activity of actinomycete species isolated from giant honey bee (*A. dorsata*) combs. *In vitro*, the crude extract of *S. ramulosus* (DSC3-6) had the highest activity against the growth of *X. oryzae* pv. *oryzae*. According to the current results, actinomycetes associated with the giant honey bee could be a good source of bioactive compounds for use in agriculture. In further studies, potent actinomycete strains will be investigated as biocontrol agents with plants under greenhouse conditions.

## ADDITIONAL INFORMATION AND DECLARATION

### Funding

This research was supported by the Department of Microbiology, Faculty of Liberal Arts and Science, Kasetsart University of the year 2019, by the Research Promotion and Technology Transfer Center (RPTTC) of the Faculty of Liberal Arts and Science, Kasetsart University Kamphaeng Sean campus, Thailand and by Grant SRIF-JRG-2562-04 from the Faculty of Science, Silpakorn University, Nakhon Pathom, Thailand. The funders had no role in study design, data collection and analysis, decision to publish, or preparation of the manuscript.

### Grant Disclosures

The following grant information was disclosed by the authors:
Department of Microbiology, Faculty of Liberal Arts and Science, Kasetsart University of the year 2019, by the Research Promotion and Technology Transfer Center (RPTTC) of the Faculty of Liberal Arts and Science, Kasetsart University Kamphaeng Sean campus, Thailand

and by Grant SRIF-JRG-2562-04 from the Faculty of Science, Silpakorn University, Nakhon Pathom, Thailand.

## Competing Interests

The authors declare there are no competing interests.

## Author Contributions

- Yaowanoot Promnuan conceived and designed the experiments, performed the experiments, analyzed the data, prepared figures and/or tables, authored or reviewed drafts of the paper, and approved the final draft.
- Saran Promsai and Sujinan Meelai conceived and designed the experiments, authored or reviewed drafts of the paper, and approved the final draft.

## Field Study Permissions

The following information was supplied relating to field study approvals (i.e., approving body and any reference numbers):

The three hive samples were collected from local villages in private areas. Verbal permission was acquired from each of the owners:

- Mr. Ton Tatiya (2 hives) 25/2 Moo2, Pong Yaeng Sub-district, Mae-Rim district, Chiang Mai Province, Thailand.

- Mr. Ma Madamun (1 hive) 93/3 Moo2, Pong Yaeng Sub-district, Mae-Rim district, Chiang Mai Province, Thailand.

## DNA Deposition

The following information was supplied regarding the deposition of DNA sequences:

The almost-complete 16S rRNA gene sequences described here are accessible via GenBank: LC536752 to LC536754.

## Data Availability

The phylogenetic trees are available in Fig. 2.

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
