# Peer review of "Antimicrobial activity of Streptomyces spp. isolated from Apis dorsata combs against some phytopathogenic bacteria"

_PeerJ, doi:10.7717/peerj.10512_

## Round 0.1 · original submission · Major Revisions

Dear Dr. Promnuan and colleagues:

Thanks for submitting your manuscript to PeerJ. I have now received three independent reviews of your work, and as you will see, the reviewers raised some concerns about the research. Despite this, these reviewers are optimistic about your work and the potential impact it will have on research studying antimicrobial activity of Streptomyces spp. isolated from the giant honey bee. Thus, I encourage you to revise your manuscript, accordingly, taking into account all of the concerns raised by the three reviewers.

Please work to clearly define your hypothesis in the Introduction. Also, please draw a clear distinction between use of Streptomyces strains themselves as biocontrol measures and the use of antibiotics isolated from Streptomyces as treatment for plant pathogens (reviewer 1).

Please explain more regarding the unpublished results (Lines 230-2 and 268-270). Consider adding these data or not mentioning the results at all.

Why are the supplemental figures identical to Figure 1?

Importantly, please ensure that an English expert has edited your revised manuscript for content and clarity.

Please also ensure that your figures and tables contain all of the information that is necessary to support your findings and observations. All work should be repeatable. All datasets must be deposited in a stable public repository prior to publication. All programs should be the updated current versions.

While the concerns of the reviewers are relatively minor, this is a major revision to ensure that the original reviewers have a chance to evaluate your responses to their concerns. There are many suggestions, which I am sure will greatly improve your manuscript once addressed.

Good luck with your revision,

-joe

Reviewer 1 ·

Basic reporting

There are several instances of incorrect English grammar in the manuscript. I have provided a list of what I observed below in the "General Comments" section.

Experimental design

There is no defined research question in the article. Instead, this article reports an observation that Streptomyces isolated from Apis dorsata honey and pollen exhibit activity against four Gram-negative plant pathogens. There is no indication as to why the authors chose to investigate the activity of bee-associated isolates against phytopathogenic bacteria.

When constructing phylogenies, the authors should use maximum-likelihood approaches, instead of neighbor-joining approaches. See the "General Comments" section for further detail.

Validity of the findings

At the end of the Introduction, the authors state: "According to data obtained in the current study, the antibacterial activity of secondary metabolites from actinomycetes could be implemented against phytopathogenic bacteria to assist in crop protection worldwide." I believe that this is an overstatement of the findings presented in the manuscript. Though it is very likely that the activity against the phytopathogenic bacteria is due to production of secondary metabolite(s), the authors have not demonstrated conclusively that their extracts contain these compounds. Further, there is no assessment of the ability of these Streptomyces to protect plants from phytopathogenic bacteria in situ.

Additional comments

In this manuscript Promnuan et al. report the isolation of twenty-five actinomycete isolates from pollen, honey, and adult workers of the bee Apis dorsata. Of these twenty-five isolates, the authors identify three Streptomyces isolates (DSC3-6, DGA3-20, and DGA8-3) that were active against four plant pathogens (Erwinia carotovora, Ralstonia solanacearum, Xanthomonas campestris pv. campestris, and Xanthomonas oryzae pv. oryzae) in a cross-streak assay. Subsequently, the authors generated crude extracts from these strains and measured their minimum inhibitory concentration (MIC) against the four plant pathogens in vitro.

Comments

1. Lines 176-7: The authors use MEGA version 6.06 and the neighbor joining (NJ) approach the generate phylogenetic trees of their Streptomyces isolates with other strains. This version of the MEGA software is from 2013. The newest version is MEGA X (Kumar et al. 2018. Molecular Biology and Evolution) version 10.1.8, which was released in April 2020. I would also recommend that the authors use maximum likelihood (ML) approaches to generate their phylogenetic trees, instead of NJ. The NJ algorithm is phenetic and does not consider evolutionary models, which are used by the ML algorithm. ML is available in MEGA X.

2. Lines 214-7: Bootstrap support is not an appropriate measure for defining the closeness of a relationship between two organisms. Without measures, such as a genome scale phylogeny or average nucleotide identity across the genome, I would be hesitant to make claims on the “closeness” of the relationship between these isolates and other strains. That said, the specific identity of the isolates does not impact the other findings reported in the manuscript.

3. Is there a link between these Streptomyces isolates from A. dorsata and the plant pathogens the authors chose to investigate? Inhibition of Gram-negative bacteria by Streptomyces is generally low. Is there specific inhibition of these plant pathogens or do these three Streptomyces isolates show more broad inhibition against other Gram-negative bacteria? How do the extract MIC values compare to other extracts from different Streptomyces?

4. Paragraph 234: The content in this paragraph would fit better in the Introduction than in the Discussion section, especially given the ending sentence: “These reports indicated that actinomycetes, especially Streptomyces, may provide a new approach for the use of actinomycetes for biocontrol in agriculture.”

Minor Comments

Line 41: Include a hyphen between “Gram” and “negative”
Line 43: Add the “e” to “seed-borne”
Line 48: Remove “the” from “…vessel causing the yellow/white…”
Line 51: Remove the “s” from the end of “causes” in “…pathogen can causes disease…”
Line 114: The sentence “The actinomycetes were isolated from three hives of A. dorsata” is redundant with the information presented under “Sample Collection”.
Lines 113-22, 128, 134: Include recipes for the bacteriological medium used in the actinomycete isolation, plant pathogen cultivation, and bioassay experiments.
Line 166: Report the centrifuge speed in × g, instead of RPM
Line 167: Replace both instances of “were” with “was”
Line 179: Why was Streptomyces thermocarboxydus DSM 44293T chosen as the outgroup?
Line 183: Include a sentence indicating that the sequences were deposited into GenBank and their accession numbers.
Line 199: Remove the word “growth” in “…growth phytopathogenic bacteria…”
Lines 209-11: Rephrase the sentence to not use the verb “blasted”
Lines 234-7: Remove “such as antibacterial and antifungal compounds” or rephrase sentence to improve clarity.
Line 237: Remove “The” from “The actinomycetes…” and start sentence with “Actinomycetes…”
Paragraph 253: There have been several recent papers assessing the large scale antimicrobial potential of insect-associated Streptomyces that have not been cited here.

Reviewer 2 ·

Basic reporting

The English is mostly clear and could be improved in several sections throughout the paper:
• Line 19 – The antimicrobial activity was not isolated so the sentence should read “The aim of this study was to investigate the antimicrobial…”
• Line 21 – The manuscript describes three media used to isolate actinomycetes and the abstract says “four different media”
• Line 81 – the phrase “and suggests” should be changed to “which suggests”
• Line 85 – the word “always” should be removed as the authors list several different places the comb could be located
• Line 86 – “and sometimes” should be changed to “or sometimes”
• Line 88 – “never been studied and reported.” should read “never been studied or reported”
• Line 99 – “bee” should be plural “bees”
• Line 195 – “show potent activity” should read “showed potent activity”
• Line 199 – “growth phytopathogenic bacteria” should read “growth of phytopathogenic bacteria”
• Line 204 – The last sentence of the paragraph starting on this line is repeated almost exactly on lines 221-223. I recommend removing the sentence on lines 221-223
• Line 208 – the word “potential” can be removed as the inhibition was apparent in the previous section
• Line 211 – “blasted” should be changed to “analyzed by BLAST”
• Line 226 – The sentence should be restructured to read “This study investigated antagonistic activity of actinomycetes isolated from the giant honey bee (A. dorsata), against phytopathogenic bacteria.
• Line 230 – “study the actinomycete” should be changed to “study an actinomycete”
• Line 231 – “activity for the growth” should be changed to “activity against the growth”
• Line 234 – 16S rDNA should be changed to 16S rRNA to be consistent with the rest of the manuscript
• Line 237 – The sentence beginning “The actinomycetes isolated” should be changed to “Actinomycetes isolated”
• Line 261 – “bee” should be made plural to “bees”
• Line 266 – “previous study reported” should be changed to “previous study we reported”
• Line 271 – “antimicrobial product for” should be changed to “antimicrobial products for”
• Line 274 – This sentence should be written “Bacterial bright leaf (BLB) disease, which is caused by X. oryzae pv. oryzae, is one of the major diseases of rice in Asian countries”
• Line 276 – the word “studied” should be removed from “in vitro studied”
• It is unclear what the authors mean by “greenhouse conditions” in line 280.

The relevant literature seems to be mostly referenced and sufficient background is provided in the introduction of the manuscript, however, the manuscript could be improved by drawing a clear distinction between use of Streptomyces strains themselves as biocontrol measures and the use of antibiotics isolated from Streptomyces as treatment for plant pathogens:
• Several times in the Discussion section it is unclear if the authors are arguing for the use of the strains they have identified as biocontrol agents themselves, as in line 252 or as sources of isolated antibiotics as in line 271 and 279. A clear distinction should be made between the two and the authors should clarify which approach they are proposing.

The article conforms to the structure suggested by PeerJ and the figures and tables are relevant to the text in the manuscript.

The results presented are relevant to the hypothesis being tested.

Experimental design

The research is presented in the manuscript is within the aims and scope of PeerJ.
The research question is very nicely set up in the introduction and the relevance of the research is made clear. It is stated in the manuscript that this is the first study to isolate Streptomyces with antibacterial activity against phytopathogenic bacteria from the giant honey bee.

Relevant controls are used in the experiments and replication was performed. Streaking the cultures from the microtiter plates onto nutrient agar plates during the MIC experiments provided extra validity to the experiment beyond simply checking for turbidity. The authors used the program MEGA version 6.06 to create their trees and perform their bioinformatic analyses.
• MEGA is now running on version 10.04, as of 2018, and is free to download. The trees and bioinformatic analyses should be performed with the most up to date version of the software.
• Additionally, the MIC values are reported in mg L-1 as opposed to the more commonly reported µg mL-1.

The methods are well described and rely on previously published protocols with minor adjustments that are described in the manuscript. The experiments could be replicated with access to the same strains.

Validity of the findings

The 16S rRNA sequences have been deposited into online databases and the experiments appear to be robust and controlled.

The conclusions require additional editing in the manuscript prior to publication.
• The three Streptomyces strains capable of inhibiting the phytopathogens were not isolated from the giant honey bee, but were rather isolated from pollen (DGA3-20 and DSC3-6) and honey (DGA8-3). The title of the manuscript and the text throughout the manuscript state that the Streptomyces strains were isolated from Apis dorsata, but this is not reflected in the data, where the single strain isolated from Apis dorsata did not have inhibitory activity against the phytopathogens. This is relevant as a 2019 Nature Communications paper by Da-Ran Kim et al. found that an endophytic Streptomyces from the strawberry rhizosphere could be detected in pollen samples of the strawberry plants. This pollen containing the Streptomyces could then be carried by pollinating bees between flowers and had the ability to protect both the strawberries as well as the pollinating bees from pathogens.

Kim, D., Cho, G., Jeon, C. et al. A mutualistic interaction between Streptomyces bacteria, strawberry plants and pollinating bees. Nat Commun 10, 4802 (2019).

The Streptomyces isolated by the authors may originate from plants that the bees were pollinating as opposed to the bees themselves. This must be addressed in the manuscript and the title of the manuscript should be altered to reflect this.

Additional comments

The authors have identified three Streptomyces strains associated with the giant honey bee and closely related to known Streptomyces species. The strains and organic extracts of the strains have antibacterial activity against four Gram-negative plant pathogens that collectively cause damage to many crop species around the world. The authors argue that the giant honey bee may be a source for the isolation of Streptomyces that produce agriculturally useful antibiotics.

·

Basic reporting

English needs to be reviewed.
The authors can use some more actual literature references.
The article has few figures to describe the work and did not present the raw data.
The results are relevant to hypotheses.
I have a few suggestions for improvement as listed below.
- I recommend modifying the Abstract, adding the background, and reduce the information described of results.
- line 63 adds the reference after the sentence "The management of plant diseases is difficult."
- In lines 80 and 99, the authors rarely mention the efficacy of Actinomycetes associated with bees which a good antimicrobial in combating phytobacterial pathogens. I suggest that you improve the description and add more details on this topic?
- line 80 and add the reference.
- line 83 The authors started the descriptions of giant bees without any association with the phytobacterial pathogens. Please, reformulated this paragraph.
- Please adding the figure of the test of the screening of antagonistic actinomycetes is a very important part of this research.

Experimental design

The idea is original and the research question is relevant and meaningful.
I have a few suggestions.
- After isolation of colonies was made morphological classification and microscopy images? Please, arrange it if not done.
- line 139-147 add the reference. What is the yield of this bioactive extraction protocol? It is important to indicate these values in the text.
- Why didn't you use bactericides, hypochlorite, formaldehyde solution, and antibiotic in MIC control?
-I suggest adding the figure of isolation of actinomycetes from A. dorsata in Supplemental files.
- Add the Evalue and Percent Identity used for the selection of the Streptomyces sequences.
- In line 207 and 209 the authors described “using 16S rDNA”, and in line 209 and 212 you use “16S rRNA”. Please, standardize the information.
- In line 209 and 212 “The almost-complete 16S rRNA gene sequences”. This information can be removed, and leave only The 16S rRNA gene sequences.
- Add in Supplemental files the figure of CLUSTAL_X alignment.
- The image of Figure 1A is a different format from the others, please, put it in the same format.
- Why the author submitted the Figure1 in Supplemental files?
- line 234 describes more studies about Streptomyces secondary metabolites with antagonistic activity against phytopathogens.
- line 237 add the reference.
- I suggested the reformulation of the Discussion topic, it is very important to associate the results obtained with those already described in the literature. Thus, the results do not make sense.

Validity of the findings

The discovery is really interesting, the results and discussion need to be improved to cooperate with the novelty described by authors.

---

## Round 0.2 · Minor Revisions

Dear Dr. Promnuan and colleagues:

Thanks for revising your manuscript. The reviewers are mostly satisfied with your revision (as am I). Great! However, there are more edits to make and suggestions to consider. Please address these ASAP so we may move towards acceptance of your work.

In particular, reviewer 2 has pointed out that the paragraph written in their previous review was a suggestion to incorporate the findings of the Nature Communications paper, Kim et al., 2019, into the discussion of this paper. You seemed to have used reviewer 2’s summary of the paper nearly verbatim in your discussion. Firstly, this does not make grammatical sense and is clearly written differently from the rest of the manuscript. Second, the direct use of reviewer 2’s summary is inappropriate. Please revise to fit the text of your own writing.

Also, please ensure that an English expert has helped with your next revision.

Good luck with the revision,

Best,

-joe

Reviewer 1 ·

Basic reporting

Overall, the English used in the manuscript is clear. I have included some suggestions that could improve clarity of a few sentences (See Line Comments, below).

The literature references are sufficient.

All data is shared.

Experimental design

no comment

Validity of the findings

All underlying data have been provided.

There are several instances in the manuscript where it is specified that the actinomycetes were isolated from A. dorsata, which is not the case. The title has been changed to reflect that most isolates were from pollen or honey, but the text needs to be updated (e.g., Lines 123, 283).

Additional comments

Promnuan et al. report the isolation of 25 actinomycetes from the surface of adult workers of the honeybee Apis dorsata, honey, and pollen. Subsequently, the isolates were tested for antimicrobial activity against four plant pathogens with three isolates, later identified as Streptomyces by 16S rDNA gene sequencing, showing activity against all the plant pathogens.

In their revision, Promnuan et al. have addressed all of my initial comments. Below I provide one additional major comment and three minor comments. In addition, I have included line comments that may improve the clarity throughout the manuscript.

Major comments

According to Table 1, only one actinomycete was isolated from a honeybee, compared to 21 isolates from pollen and 3 isolates from honey. Further, the three isolates for which antimicrobial extracts were generated, were from honey and pollen, not the honeybee. Therefore, it is not necessarily fair to state that the actinomycetes were isolated from A. dorsata (e.g., Lines 123, 283). The manuscript title was changed, but the manuscript text needs to be updated accordingly.

Minor comments

Since the scientific name of Erwinia carotovora was updated to Pectobacterium carotovorum, then the references to E. carotovora need to be updated to P. carotovorum throughout the manuscript. (e.g., lines 41, 146, 173, 177, 217, 223)

All scientific names should be spelled out fully the first time that they are used (e.g., Lines 79-81, 102, 110, 113)

Lines 125-6: It is not yet valid to say that the antibacterial activities demonstrated in this manuscript are due to secondary metabolites.

Line Comments

Lines 26, 28, 98, 139, 221, 222, 224, 266: Insert a space between mg and L-1.

Line 32: Update the abstract to reflect the use of maximum likelihood approaches instead of neighbor joining for generating phylogenetic trees.

Line 78: The abbreviation BLB has already been defined.

Line 91: What was the similarity metric for 99.6%? 16S rDNA sequence?

Lines: 92, 96, 98: These lines describe strains with the highest/maximum inhibitory activity from different studies, but do not provide context for the comparison.

Line 145: Ralstonia solanacearum should be abbreviated to R. solanacearum

Line 209, 244: Replace ‘glycerol-asparagine (ISP5) agar)’ with ‘ISP5 agar’, given that this abbreviation was used in the methods.

Lines 224-5: This sentence would fit better in the conclusion section of the manuscript, instead of in the results.

Line 228: Did the authors mean to write ‘…showed inhibition of all four phytobacterial pathogens’ instead of ‘…all three…’ as they tested four different plant pathogens? (see line 216)

Lines 248-50: This sentence is redundant with lines 244-7.

Lines 250-1: Could the authors clarify this sentence? I am not sure what they are referring to.

Line 265: Delete the word ‘the’ from ‘…showed the activity’…

Line 281: Replace ‘Bacterial bright leaf (BLB) disease…’ with ‘BLB disease’ as the abbreviation was defined at line 46. Alternatively, delete the abbreviation at all three locations (lines 46, 78, 281) and use the full name ‘Bacterial bright leaf disease’.

Reviewer 2 ·

Basic reporting

The English could be improved in several sections throughout the paper:
Line 31 - the word "sequence" should be plural "sequences"
Line 34 - The word "respectively" should be added at the end of the sentence to provide clarity
Line 229 - The authors are testing against four phytobacterial pathogens, not three
Line 275 - Change "from the in vitro screening method" to "in vitro"
Line 276 - Change "and reducing" to "and reduced"
Line 276 - "A pot experiment" is this an in situ experiment or in vivo?
Line 279 - The last sentence "Further investigation on potent actinomycete strains as biocontrol agents under greenhouse conditions" has no verb and is not a complete sentence. Additionally, it is repeated below in the conclusions section. It should be removed or edited.
Line 287 - The English in the last sentence should be altered. I believe it should read "In future studies, the potent actinomycete strains will be investigated as biocontrol agents with plants under greenhouse conditions."

Relevant literature seems to be referenced and sufficient background is provided.

The article conforms to the structure suggested by PeerJ and the figures and tables are relevant to the text in the manuscript.

The results presented are relevant to the hypothesis being tested.

Experimental design

The research presented in the manuscript is within the scope of PeerJ.

The research question is nicely set up in the introduction and the relevant research is made clear. The manuscript states that this is the first study to isolate Streptomyces with antibacterial activity against phyopathogenic bacteria from the giant honey bee.

Relevant controls are used in the experiment and replication was performed. The authors used up-to-date versions of their bioinformatic software.

The methods are well described and rely on previously published protocols with minor adjustments that are addressed in the methods section.

Validity of the findings

The underlying data has been deposited into the relevant databases.

The fact that the potent actinomycete strains were mostly isolated from pollen is addressed and this is reflected in both the discussion and the title. However, the English in the discussion section needs to be edited. The sentences beginning on line 251 and ending on line 254 are copied nearly directly from my previous review. While the sentences aim to provide context from the literature for the sentences following them, they do not make sense grammatically as they are written and should be written in the authors own words.

·

Basic reporting

The authors accepted all the suggestions made previously, improving the quality of the text.

Experimental design

I have a few comments to improve the text.

line 87 "soilborne" change to soil-borne;
In line 194 the authors wrote "After incubation, plates were observed visually. " why not use this method instead of the absorbance method?
line 203- Add the time of centrifugation;
line 206- Add a reference;
line 208 - Describe what type of sequencing was performed. And add in the accession number of the sequences obtained from the tested isolates;
line 257- I suggest that the authors make a single phylogenetic tree for the 3 isolates, so it is easier to visualize and understand the result.

Validity of the findings

No comment

---

## Round 0.3 · accepted · Accept

Dear Dr. Promnuan and colleagues:

Thanks for revising your manuscript based on the concerns raised by the reviewers. I now believe that your manuscript is suitable for publication. Congratulations! I look forward to seeing this work in print, and I anticipate it being an important resource for groups studying antimicrobial activity of Streptomyces spp. isolated from the giant honey bee. Thanks again for choosing PeerJ to publish such important work.

Best,

-joe